# Is There Financialization of Housing Prices? Empirical Evidence from Santiago de Chile

**José-Francisco Vergara-Perucich**

Centro Producción del Espacio, Universidad de Las Américas, Manuel Montt 948, Providencia 7500975, Chile; jvergara@udla.cl

**Abstract:** This paper empirically examines the influence of financial factors on housing prices in Chile, given the relevance that arises from the important housing crisis that the country is going through and the scarce amount of literature on causal relationships between price and other variables for the case under study. The hypothesis is that different financial factors have a significant influence on the price of housing, while the price of housing acts as an attractor of financial investment due to its good profitability. To conduct the statistical test, a Granger causality test is applied to a weekly data series covering the years 2009–2018. The results indicate that the causality hypothesis is plausible. On the one hand, the role of international stock market influences housing prices in Chile. Also, the Chilean Central Bank has a significant causality relationship with the housing prices. The structure of the article is organised with a second section devoted to explaining the methods and data used for the test, followed by the results analysed and discussed in the third section and closing conclusions that allow for further research and policy implications of the findings. These findings are valuable to complexify the Chilean housing policy by incorporating financial variables.

**Keywords:** financialization; housing prices; Granger Causality; Chile





## 1. Introduction

One of the most relevant explanatory vectors of the sub-prime crisis was the significant increase in mortgage lending as the main mechanism for access to housing, in a scheme characterised by the literature as the financialisation of housing (Lapavitsas 2009). Although the subprime crisis was caused by a strong credit and debt component, the social structures underpinning the economy have rather historical characteristics (Bone 2009) which have not changed much since their breakdown in 2008 (Fine 2022; Ivanova 2017; Reid 2017). Contradictorily, after the subprime crisis, the financialisation of housing continued its course, generating huge amounts of unoccupied housing in globally relevant urban centres, while reproducing the scarcity of housing tenure security for millions of families (Ferreri and Vasudevan 2019). The financialisation of housing in the Global South is partly a consequence of housing financialization processes in the Global North, through liquidation of excess liquidity through investment in fixed assets—housing—in cities in the Global South (Fernandez and Aalbers 2020).

In the case of Islamabad, Akhtar and Rashid point out the relationship between displacement, real estate development and financialization in gated communities that are based on processes of dispossession and ownership of land by financial capital with military rationality in the strategies used (Akhtar and Rashid 2021). In Mexico, the case of facilitating access to finance for low-income households to produce housing solutions generates ambivalent outcomes between banking inclusion and the conflicting rationalities of financial systems (Grubbauer 2019, 2020). On the other hand, in places in the global south where the market is underdeveloped, processes can occur in which international financial institutions create capital accumulation regimes through the creation of financial markets for housing, a case that Fauveaud has documented and researched in Cambodia

(Fauveaud 2020). In the case of Chile, the process of housing financialization has been under development for a significant time due to the entry of the capital market in 2004 to finance real estate development, along with a significant rate of bankarization of the nation (Hidalgo et al. 2014; Hidalgo Dattwyler et al. 2019; Cattaneo Pineda 2011; Vergara-Perucich and Boano 2019). In the case of Brazil, the effects of the financialization process have been identified that not only affect the increase in households with mortgage debt but also have an impact on the strategies of the real estate development industry (Fix and Arantes 2021; Sanfelici and Halbert 2014). Much of the research on housing financialization focuses on the social effects, but there is not much literature reviewing the effect of financialisation on the economy itself.

Hurn et al. investigated the interconnectedness of housing markets through international networks by looking through temporal causal relationships for possible cross-influences, which was not fully confirmed but did identify a field of research relevant to economic disciplines (Hurn et al. 2022). A study in Auckland identified causal relationships between speculative housing investment and long-term house price increases (Yang et al. 2019). Liang et al. evaluate the causal relationship between commodity prices and local house prices, identifying that there is no significant evidence of linear causality between the two, although there is significant evidence for non-linear relationships measured by the Granger causality test (Liang et al. 2021). One study indicates a significant causal relationship between stock market share prices and house prices in the UK (Bissoondeeal 2021). In line with the above, Wilhelmsson identifies a causal relationship between interest rates set by monetary policy and house prices (Wilhelmsson 2020). While there is evidence that identifies causal relationships between financial variables and house prices, the organisation of these studies within the financialization framework is not fully stated. Concerning the housing submarket in Chile, Aye (2018) conducted a Granger causality analysis to see if economic policy uncertainty affected housing prices, by grouping macroeconomic variables at the level of eight nations including Chile, but not with a specific focus on the Chilean housing market. Hong and Li conducted a Granger causality analysis between housing prices and the willingness to invest of financial actors for the case of China (Hong and Li 2020); however, there are no other articles before this one that evaluate the causal relationship from the theoretical framework of financialization with housing prices, neither for the case of Chile nor for other cases as far as this research project has been able to review. This makes the results presented here of great value to think tanks in the field of financialization, housing and real estate markets.

In addition, the Chilean case is special because of the housing market conditions characterised by very low normative regulation, privileging the action of market agents to determine where new housing units need to be produced without clear central planning for public purposes (Vergara-Perucich and Aguirre-Nuñez 2020). This market model is being transformed because of empirical evidence of its shortcomings; however, these changes are not yet structural, so the housing market in Chile offers a unique case in terms of reviewing supply, demand and price formation relationships without state regulation (Boano and Vergara-Perucich 2017; Vergara-Perucich 2019). Precisely because of the absence of state regulation, the explanatory variables on housing prices may be due to factors exogenous to the productive process of the real estate market, because macroeconomic factors impact the domestic economy without many obstacles. For example, in 2004 Leung argued that at that time the literature on property price dynamics was scarce and unsatisfactory, suggesting that it was important to move towards intertwined explanations between macroeconomic factors and house price cycles (Leung 2004). With the 2008 subprime crisis, the literature began to study house price cycles in relation to macroeconomic variables in greater depth, broadening this field of research. Liu et al., applying a Gordon growth model, indicate that there is a relatively significant covariance between growth expectations and house prices, suggesting that bubbles, i.e., price speculation, could be considered as a key driver of house prices in the Chinese case (Liu et al. 2017). In a cointegration study for the Sydney case, Al-Masum and Lee indicate that domestic market fundamentals had statistically significant

relationships with house prices in the long run (Al-Masum and Lee 2019). Complementarily, Bangura and Lee in 2022 indicated that these cointegration processes in Sydney depend on each submarket within Greater Sydney (Bangura and Lee 2022). In the case of Chile, this type of causal modelling is absent in the literature.

Brett Christophers raises some doubts about research on housing financialization because it is presented as a somewhat ambiguous concept; however, for Manuel Aalbers, this ambiguity allows for a broad disciplinary line of research on the role of financialization in housing (Aalbers 2015). For Jacobs and Manzi, some key variables for analysing housing financialization processes are related to mortgage interest rates and monetary policy, among others (Jacobs and Manzi 2020). For Sabaté, housing indirectly allows the leverage of other investments in the financial world, so that in one way or another there would be an influence on the housing market from the stock markets, a situation that is supported by the causality study developed by Irandoust for seven European nations (Sabaté 2016; Irandoust 2021). Financial markets are adaptable to economic fluctuations, but find good shelter in housing investment, generally a stable market with long-term fixed income (Harvey 1985, 2009; Harvey and Smith 2005). The process of financialization of housing is complex and can have specific implications for the social role of property by orienting it towards rental objectives, leaving the human right to housing in the hands of the laws of the market (Farha 2018).

Chile is currently facing one of its biggest housing crises in the last 50 years, with more than 80,000 families living in slums. Some 51.5% of these households say they live informally because they do not have sufficient resources to live in the city (Vergara-Perucich 2021b). For example, the quantitative housing deficit in Chile in 1998 was 497,012 units, while in 2017 it was 497,615. Although the state invested 22 billion dollars in subsidies during this period, in this same period there was a net increase of 0.12 % of the deficit (Observatorio Urbano 2022). While lower-income households find it difficult to access secure housing, the financial market generates significant profits from its real estate investment portfolio. In 2020, there were 120 financial funds with real estate assets in Chile managing USD 4742 million—where 67% of the investment sought rental income. To tackle this case, the research focuses on Santiago de Chile, its main metropolis.

This paper aims to generate an empirical examination of the influence of financial factors on housing prices in Chile, given the relevance that arises from the important housing crisis that the country is going through and the scarce amount of literature on causal relationships between price and other variables for the case under study. The hypothesis is that different financial factors have a significant influence on the price of housing, while the price of housing acts as an attractor of financial investment due to its good profitability. To conduct the statistical test, a Granger causality test (Chee-Yin and Hock-Eam 2014; Granger 1969; Gujarati and Porter 2009) is applied to weekly data series between 2008 and 2019.

The results indicate that the causality hypothesis is plausible. On the one hand, the role of international stock market influences housing prices in Chile. Also, the Chilean Central Bank has a significant causality relationship with the housing prices. The structure of the article is organised with a second section devoted to explaining the methods and data used for the test, followed by the results analysed and discussed in the third section and closing with conclusions that allow for further research and policy implications of the findings.

## 2. Methodology

To study the Chilean case, we specifically review the submarket of Greater Santiago, the nation's main metropolitan area that concentrates 79% of the housing market supply and 40% of the national population. Wu and Sharma point out the advantages of analysing a submarket based on the presence of continuous and interdependent spatial divisions that allow for a comparative analysis of spatio-temporal changes in the housing markets under investigation (Wu and Sharma 2012). The case of Santiago de Chile, in addition

to concentrating most of the national housing market, is composed of 32 interdependent communes with high levels of socioeconomic segregation and spatial division of functions (Correa-Parra et al. 2020) that allow us to meet the criteria suggested by Wu and Sharma to enrich housing market studies.

Santiago is a highly segregated city (Sabatini et al. 2020; Toro and Orozco 2018) so there are significant disparities in the different areas of the metropolis, a situation that has an impact on the housing market for emerging economies (Randolph and Tice 2014; Wang and Lee 2022). In addition, Chile is a nation with very unequal cities (Vergara-Perucich 2021a), so the consideration of Santiago implied defining this metropolitan space as a specific submarket. In several previous studies, the spatial qualities and organisations of housing markets are not considered methodologically and therefore can generate results that are difficult to generalize as indicated by Costello et al. (2011). In this case, being aware of the high segregation of the housing market in Greater Santiago and the inequality among Chilean cities, an ad-hoc methodological design was considered necessary, given that the macroeconomic variables included in this study would possibly not have the same effect in smaller cities or cities dependent on other productive activities within the same nation. For example, Idrovo-Aguirre and Contreras-Reyes have identified that cities in northern Chile are more sensitive to changes in the price of copper, given that these cities are highly dependent on this commodity (Idrovo-Aguirre and Contreras-Reyes 2021). In the case of Santiago, the main economic dependence is on financial services.

The model uses a set of individualised time series for the group of variables summarised in Table 1. The criterion for including these variables arises from the discussion in the introduction above. First, as Bissoondeeal (2021) argued, stock markets can influence house prices. For this reason, the IPSA of the Santiago stock exchange is incorporated into the analysis, given that it is the financial market performance indicator for the national territory, where real estate investment groups, developers and financial institutions themselves participate. A second factor mentioned in the literature by Wilhelmsson (2020) is the monetary policy rate defined, in the case of Chile, by the Central Bank. Associated with this strategy, the amount of money in circulation is also a strategy defined by the Central Bank in the case of Chile, so it has been decided to incorporate this factor into the causality assessment to review whether the Central Bank influences housing prices and how many periods can be identified for these effects to be reflected in prices. Related to the above, in Chile, the Central Bank tends to set its monetary policy seeking to protect the market from inflation, so inflation is also incorporated as a variable to be reviewed concerning house prices. Following the findings of Irandoust (2021) and Sabaté (2016), this is because financial markets are global and Chile is a nation widely open to international markets, with free trade agreements with 65 nations. Among the nations with the largest trade relations are China, the United States and the United Kingdom. For this reason, it was decided to incorporate the performance indicators of the main stock exchanges of these nations to see if they have any effect on the price of housing in Chile.

**Table 1.** Summary of descriptive variables (ln).

| Variables | Obs | Min. | Max. | Mean | St. Dv. | Kurtosis |
|---|---|---|---|---|---|---|
| Housing Prices (ln) | 472,000 | 2020 | 5060 | 3658 | 0.434 | 1.554 |
| IPSA (Santiago Blue Chip Index) (ln) | 472,000 | 7810 | 8670 | 8341 | 0.155 | 1.304 |
| Monetary Policy Rate Chilean Central Bank (ln) | 472,000 | −0.690 | 2090 | 1157 | 0.554 | 4.419 |
| Cash Circulation (ln) | 472,000 | 13,770 | 14,820 | 14,414 | 0.292 | −1.145 |
| Inflation (ln) | 472,000 | 9940 | 10,220 | 10,080 | 0.085 | −1.381 |
| Mortgage Interest Rate (ln) | 472,000 | 1150 | 1800 | 1378 | 0.119 | 0.395 |
| CSI 300 (ln) | 472,000 | 7560 | 8570 | 8004 | 0.206 | −0.808 |
| FTSE 100 (ln) | 472,000 | 8260 | 8960 | 8743 | 0.135 | 1.010 |
| Dow Jones (ln) | 472,000 | 8900 | 10,190 | 9646 | 0.285 | −0.487 |

In this research, it has been decided to work with weekly data by moving averages after cleaning the daily database of outliers and missing data. It has also been sought to clean the

data leaving only observations from Monday to Friday. This has been applied since some explanatory variables did not have daily series because they did not incorporate information from weekends and holidays. To unify the sample, this strategy of normalisation by weeks is carried out, resulting in an initial total of 520 weeks of continuous-time series analysis. Finally, the database is smoothed by moving averages every three weeks, as suggested by Hamilton (2018).

The initial time series runs from September 2009 to September 2019. These dates arise from two main criteria: availability of detailed data for all the chosen variable series and because in September 2009 the effect of the subprime crisis had already subsided in Chile, and in October 2019 Chile enters a process of deep social crisis that distorts the housing market as a result of a series of riots and stoppages of productive activities at different levels (Arias-Loyola 2021). By using time series with weekly observations, a high-frequency data set is achieved, which reduces the statistical noise that can be generated by lower frequency series that do not recognise the volatility of financial markets or investment risk (Lee et al. 2018), which for real estate analysis offers a greater ability to accurately study long-term variations (Cotter and Stevenson 2008). To improve the stationarity of the sample, a logarithmic conversion is performed for each variable as shown in Figure 1, which also shows the behaviour of the time series analysis variables in their natural logarithmic form.

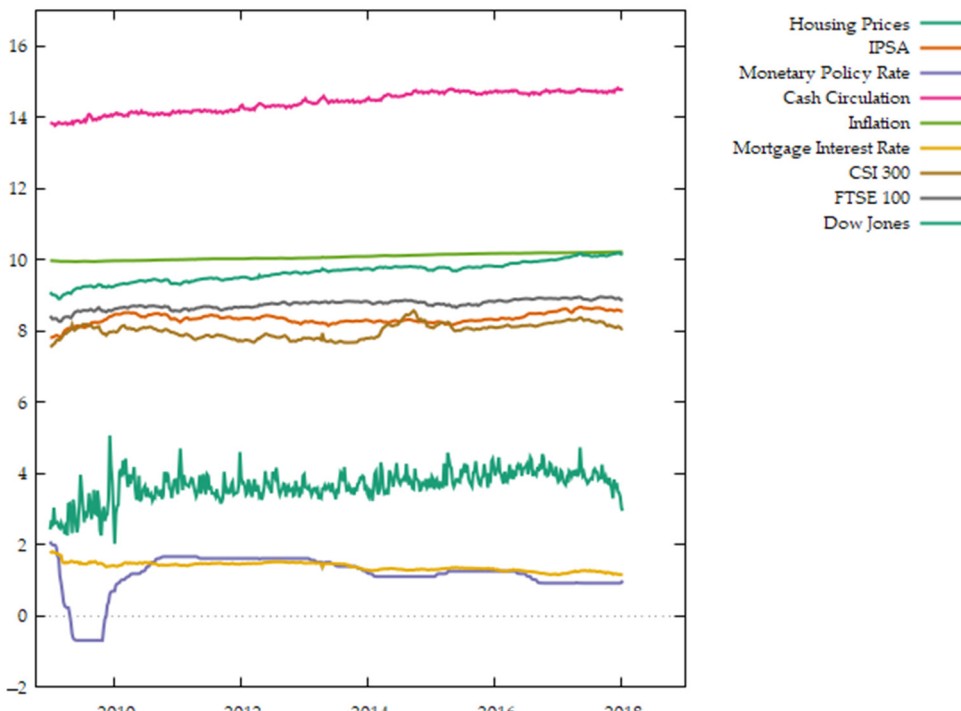

**Figure 1.** Ln of variables of the model.

To check the causal relationships of these variables that may affect house prices, a VAR based Granger causality test (Granger 1969) is used. This analysis technique is simple and tremendously useful for this purpose, where with the lagged values of a variable $x_t$ it is possible to check whether these values contribute to explaining another variable $y_t$, in the simplified notion that $x_t$ Granger causes $y_t$, or: $x_t \rightarrow y_t$ (Fernandois et al. 2020). In this case, the Granger test is useful since it seeks to check the short-run relationship between variables. Because this technique is based on a systematic review of relationships in the presence of lagged variables, the optimal lag order for the model must be defined first. For this purpose, a selection of the order of lags is applied. To identify the lags, a calculation is made based on the following abbreviated equation:

$$A(L)\ y_t = Bx_t + \varepsilon_t \tag{1}$$

where A(L) is a polynomial lag matrix, $x_t$ represents a constant exogenous variable and t is a vector. As indicated in Table 2, lag 6 by AIC and FPE is the most appropriate, while LR indicates that lag 15 is the most appropriate. SC and HQ indicate that it is better to work with lag 1. Since this exercise is an exploratory review, it was decided to apply 15 lags, as this is the result that will offer the largest number of causal relationships to analyse.

**Table 2.** Lag order selection by each criterion.

| Lag | LogL | LR | FPE | AIC | SC | HQ |
|-----|------|-----|-----|-----|-----|-----|
| 0 | 10,906.55 | NA | $8.32 \times 10^{-33}$ | −48.32617 | −48.24412 | −48.29383 |
| 1 | 11,195.36 | 564.8127 | $3.31 \times 10^{-33}$ | −49.24772 | −48.42725 * | −48.92437 * |
| 2 | 11,288.50 | 178.4303 | $3.14 \times 10^{-33}$ | −49.30155 | −47.74266 | −48.68719 |
| 3 | 11,375.56 | 163.3013 | $3.06 \times 10^{-33}$ | −49.32840 | −47.03109 | −48.42303 |
| 4 | 11,458.73 | 152.7111 | $3.03 \times 10^{-33}$ | −49.33807 | −46.30233 | −48.14168 |
| 5 | 11,537.39 | 141.2606 | $3.07 \times 10^{-33}$ | −49.32766 | −45.55350 | −47.84026 |
| 6 | 11,639.13 | 178.6770 | $2.82 \times 10^{-33}$ * | −49.41966 * | −49.0708 | −47.64125 |
| 7 | 11,714.53 | 129.3867 | $2.91 \times 10^{-33}$ | −49.39479 | −44.14378 | −47.32537 |
| 8 | 11,766.53 | 87.17210 | $3.33 \times 10^{-33}$ | −49.26621 | −43.27677 | −46.90576 |
| 9 | 11,833.97 | 110.3534 | $3.58 \times 10^{-33}$ | −49.20606 | −42.47821 | −46.55461 |
| 10 | 11,909.86 | 121.1604 | $3.71 \times 10^{-33}$ | −49.18342 | −41.71714 | −46.24095 |
| 11 | 11,966.63 | 88.35669 | $4.20 \times 10^{-33}$ | −49.07595 | −40.87124 | −45.84246 |
| 12 | 12,029.92 | 96.00096 | $4.64 \times 10^{-33}$ | −48.99745 | −40.05432 | −45.47295 |
| 13 | 12,086.15 | 83.02446 | $5.31 \times 10^{-33}$ | −48.88757 | −39.20602 | −45.07206 |
| 14 | 12,139.93 | 77.27138 | $6.16 \times 10^{-33}$ | −48.76686 | −38.34689 | −44.66034 |
| 15 | 12,214.02 | 103.5024* | $6.57 \times 10^{-33}$ | −48.73624 | −37.57784 | −44.33870 |
| 16 | 12,286.64 | 98.54849 | $7.09 \times 10^{-33}$ | −48.69909 | −36.80227 | −44.01054 |
| 17 | 12,359.48 | 95.93558 | $7.68 \times 10^{-33}$ | −48.66290 | −36.02766 | −43.68334 |
| 18 | 12,432.43 | 93.16378 | $8.37 \times 10^{-33}$ | −48.62719 | −35.25352 | −43.35661 |
| 19 | 12,508.79 | 94.47243 | $9.05 \times 10^{-33}$ | −48.60660 | −34.49451 | −43.04501 |
| 20 | 12,584.95 | 91.19765 | $9.87 \times 10^{-33}$ | −48.58516 | −33.73465 | −42.73256 |

* indicates lag order selected by the criterion. LR: sequential modified LR test statistic (each test at 5% level). FPE: Final prediction error. AIC: Akaike information criterion. SC: Schwarz information criterion. HQ: Hannan-Quinn information criterion.

Given the lags, it was checked to ensure that the difference and logarithmic transformations applied to the variables ensure the stationarity of the sample. For this purpose, an augmented Dickey-Fuller test is applied from 15 lags downwards. The result of this test can be seen in Table 3, where it is shown that all of the *p*-values are below 0.05, and in Figure 2 the behaviour of the variables over time can be seen.

To confirm the robustness of the selected dataset after transformation, for the order of 15 lags to be reviewed, an Engle-Granger cointegration test is applied which seeks to identify whether a pair of variables are cointegrated by analysing the residuals of regression between variables to identify by Augmented Dickey-Fuller test whether a unit root exists. Table 4 indicates the results confirming the robustness of the variables to applying a Granger causality test.

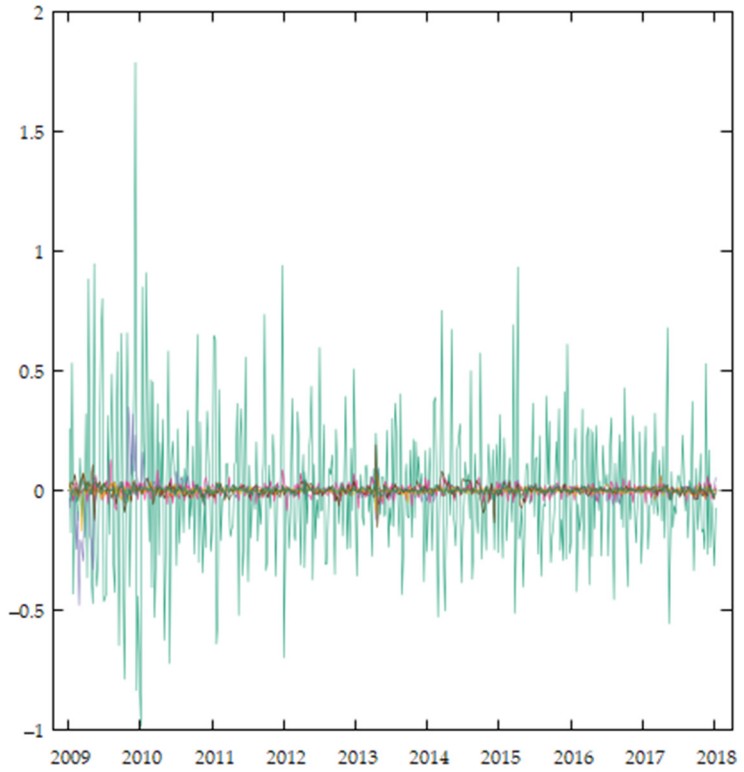

**Figure 2.** Variables after ln and transformation by 1 difference.

**Table 3.** Augmented Dickey-Fuller Test on variable ln with 1 difference by AIC criterion.

| Variable | *p*-Value | Lags | Obs |
|---|---|---|---|
| Housing Prices | $5.14 \times 10^{-20}$ | 15 | 470 |
| IPSA | $4.11 \times 10^{-27}$ | 15 | 470 |
| Monetary Policy Rate | $1.25 \times 10^{-9}$ | 15 | 470 |
| Cash Circulation | $1.72 \times 10^{-11}$ | 15 | 470 |
| Inflation | $6.25 \times 10^{-17}$ | 15 | 470 |
| Mortgage Interest Rate | $3.80 \times 10^{-12}$ | 15 | 470 |
| CSI 300 | $1.99 \times 10^{-38}$ | 15 | 470 |
| FTSE 100 | $6.04 \times 10^{-29}$ | 15 | 470 |
| Dow Jones | $1.22 \times 10^{-36}$ | 15 | 470 |

**Table 4.** Engle-Granger cointegration test results.

| Variables on Housing Prices | Lags | Engle-Granger Coefficient | *p*-Value |
|---|---|---|---|
| IPSA | 15 | −7.1113275 | 0.01 |
| Monetary Policy Rate | 15 | −6.8507553 | 0.01 |
| Cash Circulation | 15 | −7.6780385 | 0.01 |
| Inflation | 15 | −7.5782503 | 0.01 |
| Mortgage Interest Rate | 15 | −7.6728005 | 0.01 |
| CSI 300 | 15 | −7.7215235 | 0.01 |
| FTSE 100 | 15 | −7.8595879 | 0.01 |
| Dow Jones | 15 | −7.6870395 | 0.01 |

Finally, to check that the data set does not present autocorrelation among its variables, a Durbin-Watson check is performed. The reference value for no autocorrelation between variables should be between 1.5 and 2.0. In this case, the Durbin-Watson test gave a value of 1.979 for the 472 observations in its logarithmic form with an input lag. Once the data are prepared for analysis, a Granger causality test will be performed. A series of bivariate regressions between variables is developed for each lag, as follows, for all possible combinations between $(x, y)$:

1. $y_t = \alpha_0 + \alpha_1 y_{t-1} \ldots + \alpha_1 y_{t-1} + B_1 x_{t-1} \ldots B_i x_{-i} + \in_t$      (2)

2. $x_t = \alpha_0 + \alpha_1 x_{t-1} \ldots + \alpha_1 x_{t-1} + B_1 y_{t-1} \ldots B_i y_{-i} + u_t$      (3)

In the analysis, the null hypothesis is that x does not Granger-cause y in the first regression and y does not Granger-cause x in the second regression. Thus, causality can be unidirectional or bidirectional, or there can simply be no causal relationship between the variables.

## 3. Results

The results of the Granger causality test for the relationship between house prices and financial factors are recorded in Table 5 as a summary. It is important to note that the Granger causality test allows us to see how one variable affects another from the analysis over time, but does not indicate which specific mechanisms produce such predictability (Fernandois et al. 2020) so that the interpretation of the empirical evidence needs to situate such analysis in the theoretical understanding of the variables tested. The vast majority of the variables tested show a statistically significant Granger causal relationship on house prices at the first lag, i.e., one week later. Particularly significant is the Mortgage Interest Rate variable, at 0.28%. In other words, changes in the mortgage interest rate have a direct and rapid causal impact on house prices. In this sense, the thesis that abounds in the financialization literature on the relationship between financial instruments for house purchase and the price of housing is fully satisfied. The second Granger causal relationship in average order of significance was that the UK FTSE100 index, with significance at the 0.67% level, the Dow Jones index, with significance at the 2.5% level, and the CSI300 in some lags, have an impact on house prices in Chile. This finding lends validity to the thesis of Irandoust (2021) and Sabaté (2016), that in the face of an international opening of the market there may be cross-influence between foreign stock exchanges and local prices. Chile's exposure to international stock market fluctuations is based on an economy open to the world. Also, the Santiago de Chile stock market measured by the Selective Stock Price Index, IPSA, presented statistical significance for the first lag for the Granger causal relationships on housing prices, a situation that would confirm that in Chile there is a causal relationship between financial variables and housing prices. All of these aforementioned relationships are univariate, i.e., the causal relationships are on house prices and not in both directions. In this study, the only bivariate result with adequate statistical significance was between house prices and the Central Bank of Chile's monetary policy rate and vice versa. This is interesting given that house price changes can be said to cause the Central Bank to react to monetary policy. However, for this analysis, we incorporated two factors that are highly incidental to the Central Bank of Chile: cash circulation and inflation. The latter two variables presented univariate causal relationships on house prices. That is, the Central Bank is more likely to react to these factors on house price changes. What is valuable about this finding is that house prices in Chile have a Granger causal relationship with Central Bank decisions, a finding that is absent in the literature on house prices in Chile.

**Table 5.** Summary of Granger causality test results.

| Lags | Null Hypothesis: | Cash Circulation Does Not Granger Cause Housing Prices | CSI 300 Does Not Granger Cause Housing Prices | Dow Jones Does Not Granger Cause Housing Prices | FTSE 100 Does Not Granger Cause Housing Prices | Mortgage Interest Rate Does Not Granger Cause Housing Prices | IPSA Does Not Granger Cause Housing Prices | Monetary Policy Rate Does Not Granger Cause Housing Prices | Inflation Does Not Granger Cause Housing Prices |
|---|---|---|---|---|---|---|---|---|---|
| 15 | F-Statistic | 1.66765 | 1.31067 | 1.85615 | 2.31413 | 1.91560 | 1.62597 | 4.23989 | 0.86258 |
|    | Prob. | 0.0545 | 0.1915 | 0.0258 | 0.0035 | 0.0202 | 0.0638 | 0.0000002 | 0.607 |
| 14 | F-Statistic | 1.21990 | 1.23327 | 1.79310 | 2.37369 | 2.13138 | 1.57166 | 4.15127 | 0.84689 |
|    | Prob. | 0.2572 | 0.2476 | 0.0373 | 0.0035 | 0.0097 | 0.0838 | 0.0000008 | 0.6177 |
| 13 | F-Statistic | 1.33299 | 1.24443 | 1.90314 | 2.51838 | 2.24169 | 1.38144 | 4.16712 | 0.82436 |
|    | Prob. | 0.1901 | 0.2447 | 0.028 | 0.0024 | 0.0076 | 0.1646 | 0.000002 | 0.6343 |
| 12 | F-Statistic | 1.28750 | 1.16250 | 2.32321 | 3.05399 | 2.57383 | 1.37502 | 4.72740 | 0.84739 |
|    | Prob. | 0.2226 | 0.3079 | 0.0069 | 0.0004 | 0.0026 | 0.1745 | 0.0000003 | 0.6013 |
| 11 | F-Statistic | 1.23002 | 1.13423 | 2.61586 | 3.37394 | 2.88455 | 1.20657 | 5.14049 | 0.95678 |
|    | Prob. | 0.2642 | 0.3324 | 0.0031 | 0.0002 | 0.0011 | 0.2799 | 0.0000001 | 0.4855 |
| 10 | F-Statistic | 1.48642 | 1.05652 | 1.54485 | 2.28446 | 3.24311 | 1.17396 | 3.56911 | 1.14528 |
|    | Prob. | 0.1414 | 0.3949 | 0.1208 | 0.0129 | 0.0005 | 0.3063 | 0.0001 | 0.3266 |
| 9 | F-Statistic | 1.68988 | 1.03540 | 1.82947 | 2.36826 | 3.89792 | 1.25146 | 4.15348 | 1.33111 |
|   | Prob. | 0.089 | 0.4105 | 0.061 | 0.0127 | 0.00009 | 0.2616 | 0.00004 | 0.2182 |
| 8 | F-Statistic | 1.95529 | 1.28220 | 1.80430 | 1.98247 | 3.78079 | 1.49112 | 4.83081 | 1.45661 |
|   | Prob. | 0.0505 | 0.2506 | 0.0743 | 0.0471 | 0.0003 | 0.158 | 0.00001 | 0.1709 |
| 7 | F-Statistic | 1.88608 | 1.52580 | 2.42361 | 2.58599 | 3.75622 | 1.54800 | 3.54258 | 1.88767 |
|   | Prob. | 0.0701 | 0.1564 | 0.0191 | 0.0127 | 0.0006 | 0.1491 | 0.001 | 0.0698 |
| 6 | F-Statistic | 2.54386 | 1.77635 | 3.11327 | 3.31692 | 4.56755 | 1.58742 | 2.82957 | 2.27513 |
|   | Prob. | 0.0197 | 0.1022 | 0.0053 | 0.0033 | 0.0002 | 0.149 | 0.0103 | 0.0356 |
| 5 | F-Statistic | 4.09431 | 1.72642 | 3.86161 | 4.18497 | 4.69034 | 1.99215 | 3.62522 | 3.15170 |
|   | Prob. | 0.0012 | 0.1271 | 0.0019 | 0.001 | 0.0003 | 0.0785 | 0.0032 | 0.0083 |
| 4 | F-Statistic | 6.39456 | 1.95411 | 5.98701 | 6.23305 | 6.50166 | 2.92652 | 4.11817 | 5.04417 |
|   | Prob. | 0.00005 | 0.1005 | 0.0001 | 0.00007 | 0.00004 | 0.0207 | 0.0027 | 0.0005 |
| 3 | F-Statistic | 9.66062 | 2.34576 | 8.54242 | 8.97276 | 9.88393 | 3.50615 | 5.43691 | 8.03374 |
|   | Prob. | 0.000003 | 0.0722 | 0.00002 | 0.000009 | 0.000003 | 0.0154 | 0.0011 | 0.00003 |
| 2 | F-Statistic | 20.0068 | 3.92675 | 17.1855 | 17.7262 | 14.3311 | 7.67253 | 3.57453 | 16.4387 |
|   | Prob. | 0.000000005 | 0.0204 | 0.00000006 | 0.00000004 | 0.0000009 | 0.0005 | 0.0288 | 0.0000001 |
| 1 | F-Statistic | 40.9455 | 7.14528 | 34.8067 | 36.5728 | 30.8936 | 16.8868 | 4.81535 | 33.4936 |
|   | Prob. | $4 \times 10^{-10}$ | 0.0078 | 0.000000007 | 0.000000003 | 0.00000005 | 0.00005 | 0.0287 | 0.00000001 |

Figure 3 represents the Granger causal relationships between the variables with the highest statistical significance for each lag. This graph allows us to see synthetically whether some of the variables are consistent in their influence on house prices for the 15 lags studied. The consistent influence of the monetary policy rate, the mortgage interest rate, FTSE100 and the influence of house prices on the monetary policy rate, as a bivariate relationship, is observed for the 15 lags. On the other hand, the weight of inflation on house prices is diluted over the weeks, a similar situation with the circulation of money. The influence of the processes known as part of financialization is causally relevant to house prices in the case of Chile.

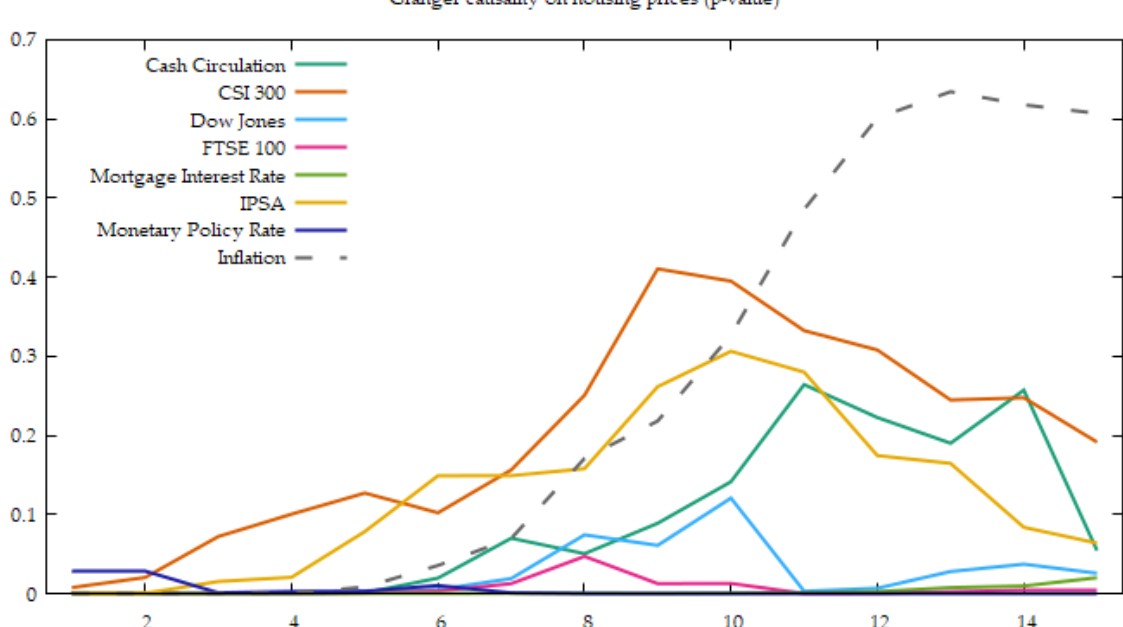

**Figure 3.** Synthetic chart of statistic value of each significant causal variable analysed.

## 4. Discussion

The results have provided a series of valuable findings to understand the phenomenon of housing financialization in the particular case of Chile. First of all, we find that the observation of Lapavitsas (2009) holds true, i.e., the mortgage interest rate is a key component in understanding housing price formation. The observed causal impact is almost direct, given that it occurs at the first lag. Second, it is valuable to see that there is a causal relationship between the UK FTSE100 and the US Dow Jones, which would allow us to advance in a line of research on the relationship between international investors and house purchases in Chile, understanding the real estate investment process as an international system, as indicated by Fernandez and Aalbers (2020). These variables are the most statistically significant and therefore it can be argued that from the Granger causality tests, the mortgage interest rate, FTSE100 and Dow Jones are the factors that best help predict variations in the price of housing in Santiago de Chile.

In the second order of statistical significance, a causal relationship appears between house prices and the Central Bank's monetary policy rate. In this sense, it can be argued that the Central Bank of Chile reacts to house prices by altering rates. In part, this is a sign of a monetary institution attentive to significant market fluctuations, especially housing prices, which is a factor that directly affects the household economy. To some extent, this finding is consistent with that of Wilhelmsson (2020), but indicates that the central bank reacts to changes in house prices. On the other hand, in lag 11, the central bank's policy rate does affect house prices in a bidirectional Granger causal relationship. That is, based on these data and in a synthetic way, when house prices rise, the Central Bank reacts, but the effects of this reaction take 11 periods to affect house prices again.

The Santiago Stock Exchange Blue Chip Index has a causal Granger effect on house prices for the first Lag, thus supporting the thesis of Bissoondeeal (2021) who made a similar finding for the UK case. In particular, the evidence presented above suggests that there is a process of financialization of house prices, where both lending rates and the financial market can influence its shape.

These results converse with similar findings in the Sydney case by Al-Masum and Lee (2019), who found causal relationships measured by cointegration between house prices and macroeconomic variables such as unemployment, GDP and household income. However, Al-Masum and Lee's methodology is different, so these results are valuable,

given that two different cases with different methodologies give validity to the causal relationship between macroeconomic variables and house prices.

These findings must be highlighted for public policy. On the one hand, we should not forget recent financial history, wherein in 2008 the process of housing financialization unleashed the devastating subprime crisis that left the world in turmoil for years. This crisis did not affect Chile at the time due to a strong fiscal policy, however, in those years, the process of financialization was not as strong as it is now. The data indicate that the Central Bank of Chile can monitor and react to house price shocks, but also the influence of stock markets on this factor can generate major problems. In Chile, there is a major housing crisis with a huge housing deficit. The evidence presented here indicates that it is pertinent to incorporate financial factors into the evaluation and design of housing policies to reduce the housing deficit, given that both institutions and financial markets are actively operating in the housing market.

## 5. Conclusions

In this paper, we use the Granger causality test to review whether the rather theoretical observations in the literature on housing financialization have empirical support for the case of Chile between 2009 and 2018. The main findings indicate that the theory of financialization presents empirical causal relationships that justify research on the relationship between financial institutions and housing affordability in the case of Chile. It is interesting to note the finding that gives importance to international stock markets, showing that international stock exchanges such as the FTSE 100 in the UK, Dow Jones in the US or the CSI 300 in China influence housing prices. Building on this specific finding, further research may explore the rationale behind these causal relationships. On the one hand, there may be international funds from these nations that invest in real estate assets in Chile, which could causally influence house prices, but in this research, such details have not been investigated. Another possible investigation is to check whether there are products necessary for the production of housing that come from these markets and that could contribute to analysing housing prices that are more complex. Then, this finding also leads to the question of whether other stock exchanges have a causal influence on housing prices in Chile.

Another valuable finding of this research is the empirical evidence of the role of the Central Bank of Chile on housing prices. Whether by setting the monetary policy rate, defining the currency strategy or acting against inflation, the Central Bank influences the price of housing and thus the ease of access of households to secure tenure from an economic perspective. Since housing is a good without substitutes, the Central Bank's cross-sectoral role in defining its monetary strategies should take these findings into account to ensure that households do not face difficulties in accessing housing. The influence of the mortgage interest rate on housing prices is finally tested, although this is a somewhat less surprising result given that one variable is designed to allow social access to the other.

The study has some limitations. Firstly, it focuses on the case of Greater Santiago, which although it concentrates a large part of the Chilean housing market, at the level of other cities the results could vary. That is, financialization processes could affect medium-sized cities and regional capitals differently. However, data at the level of these cities are scarcer and at the moment more difficult to model. It is a pending task for further studies to run these causality models at the national level. Another limitation is that there are other explanatory factors on housing prices that are not macroeconomic but that could help to make the analysis more complex, such as the price of construction materials, land prices or the socio-economic characteristics of the neighbourhood. Incorporating these variables could allow us to identify whether financial and macroeconomic factors have a greater statistical weight on house prices than other fundamentals identified in the literature.

For future studies, the data used are suitable for developing other types of time series analysis, such as ARIMA models or especially analysis with VAR models. The latter would allow the design of predictive scenarios for the variables under study with greater precision than can be achieved by employing hedonic analysis or multiple regressions.

Thus, to ensure and promote secure access to housing for households in the case of Chile, housing policy must integrate financial factors as a key structural variable in its design. The mere production of housing or the provision of subsidies will not be able to generate adequate solutions to the problem of secure housing tenure until financialization is incorporated as a causal vector with sufficient statistical significance to influence housing prices. In a free-market, property-oriented society where the housing market controls the processes of housing production and allocation, understanding housing price formation is an unavoidable public policy issue.

**Funding:** This research received no external funding and the APC was funded by Dirección de Investigación of Universidad de Las Américas.

**Institutional Review Board Statement:** Not applicable.

**Informed Consent Statement:** Not applicable.

**Data Availability Statement:** Please, find the data under a CC0 License at: https://github.com/franciscovergarap/fvergara_economies2022, accessed on 2 April 2022.

**Conflicts of Interest:** The author declares that they have no conflict of interest.

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
