# Peer review of "Is There Financialization of Housing Prices? Empirical Evidence from Santiago de Chile"

_economies, doi:10.3390/economies10060125_

Round 1

Reviewer 1 Report

This paper examines the causal relationship between macroeconomic financial factors and housing prices in Chile over the study period 2009-2018 with a Granger causality test. The results show that Chilean Central bank rate has a significant causality relationship with housing prices in Chile. Importantly, this highlights different financial factors have a significant influence on housing prices, suggesting that there is a causality relationship between housing prices and financial factors in Chile. This is an interesting study. The paper has its merit and potential. However, I’m afraid that the paper needs a major restructure before the paper is publishable. See the following comments:

Major comments:

  1. The motivation of this study can be further improved. We should motivate this study from the research gap in the literature. As such, the authors should discuss whether this is the first study to examine this topic. Is it the first study to examine the Chilean housing market? This should be discussed. Further, extra information is required to justify a dedicated study of this kind is crucial. Importantly, the uniqueness of the Chilean housing market should be discussed. How the market differs from other markets (e.g. developed markets)? Can the experiences from the developed markets be applied into the Chilean market directly? If not, then this can reinforce the importance of a study of this kind. I believe the authors should discuss this further.
  2. This can be done through an inclusive literature review. Extensive studies have been devoted to examining how macroeconomic variables can be used to explain housing prices. Unfortunately, the findings are somewhat mixed. I’m afraid that the authors did not attempt to do a proper literature to identify the research gap. Please note that this is not somewhat new. Many studies consider a number of variables or market fundamentals such as GDP, interest rate, stocks etc.  See the following examples. There are extensive studies that can be found from the literature.

Leung, C. (2004). Macroeconomics and housing: a review of the literature. Journal of Housing Economics13(4), 249-267.

Hui, E, C, M & Yue, S 2006, Housing price bubble in Hong Kong, Beijing and Shanghai: a comparative study, Journal of Real Estate Financial Economics, vol. 33, pp 301-302

Al-Masum, M.A. and Lee, C.L. (2020), "Modelling housing prices and market fundamentals: evidence from the Sydney housing market", International Journal of Housing Markets and Analysis. Vol. 12 No. 4, pp. 746-762. https://doi.org/10.1108/IJHMA-10-2018-0082

Liu, R., Hui, E. C. M., Lv, J., & Chen, Y. (2017). What drives housing markets: fundamentals or bubbles?. The journal of real estate finance and economics, 55(4), 395-415.

Bangura, M., & Lee, C. L. (2022). Housing price bubbles in Greater Sydney: evidence from a submarket analysis. Housing Studies, 37(1), 143-178.

3. The discussion of the variable selection is scant. No literature has been provided to justify the use of these variables to examine the causality relationship between housing prices and financial factors.  Why these variables have been selected? What are the theoretical frameworks can be used to justify the use of these variables?

4. In addition, the use of Santiago de Chile as the dataset for this study. I am wondering whether the use of the disaggregated data. This is a housing submarket approach instead of national level. The housing submarkets should also be discussed. Again, extensive studies of housing submarkets have not been included in the literature.

The methodology should be discussed further. The first issue should be the use of weekly data. This is an advantage of this study by using a high-frequency data. This should be discussed more clearly why the authors use the weekly data instead of other frequency. Although this is an advantage, the authors should discuss whether some data (e.g. central bank interest rate) fluctuate so frequently (weekly). Some studies have discussed the use of daily or weekly data in housing.

5. The stationary test is up to 15 lags. This equals to a lag of 3-4 months. Some explanation is required why 15 lags were selected. In addition, the authors should consider KPSS as an alternative stationary test to confirm the stationary of the variables as these are so critical steps for any Granger causality test.

6. The results can be further discussed. This can be expanded. One of enhancing the discussion is to refer the results to the previous empirical findings. For instance, mortgage interest rate causality results in this study. Is it like previous findings from Al-Masum and Lee (2020) in IJHMA and Liu et al. (2017) in JREFE? If not, then why not. This could be attributed to the importance of international evidence.

Minor comments

1) Proof reading is required. Numerous editorial mistakes have been identified.

2) Table 4- please use the granger causality coefficient as well.

3) Please follow the recommended format by Economies. This is crucial to ensure that the paper follows the recommended formats.

Author Response

RESPONSE TO REVIEWERS ECONOMIES

PAPER: Is there Financialization of Housing Prices? Empirical Evidence from Santiago de Chile

economies-1692374

Thanks to the editors of Economies for allowing me to amend the paper following the suggestions of reviewers. I did a comprehensive revision of the article, by considering the valuable contributions of reviewers. Both provided substantial observations to the paper giving precise details on how to improve it. This make it easier to address the revisions required and to improve it.

Based on these revisions, I understand the point of both reviewers was to request deeper reflections taking advantage of the data produced in the research process. Although these revisions demanded more work and re-write entire paragraphs and even new calculations, I believe the result is much better than in the original version.

Here are the responses for each requirement and/or comments made by the reviewers.

REVIEWER 1

The motivation of this study can be further improved. We should motivate this study from the research gap in the literature. As such, the authors should discuss whether this is the first study to examine this topic. Is it the first study to examine the Chilean housing market? This should be discussed. Further, extra information is required to justify a dedicated study of this kind is crucial. Importantly, the uniqueness of the Chilean housing market should be discussed. How the market differs from other markets (e.g. developed markets)? Can the experiences from the developed markets be applied into the Chilean market directly? If not, then this can reinforce the importance of a study of this kind. I believe the authors should discuss this further. This can be done through an inclusive literature review. Extensive studies have been devoted to examining how macroeconomic variables can be used to explain housing prices. Unfortunately, the findings are somewhat mixed. I’m afraid that the authors did not attempt to do a proper literature to identify the research gap. Please note that this is not somewhat new. Many studies consider a number of variables or market fundamentals such as GDP, interest rate, stocks etc.  See the following examples. There are extensive studies that can be found from the literature.

Response: Thank you for your comments and input. The following addenda have been made:

In addition, the Chilean case is special because of the housing market conditions characterised by very low normative regulation, privileging the action of market agents to determine where new housing units need to be produced without clear central planning for public purposes (Vergara-Perucich & Aguirre-Nuñez, 2019). This market model is being transformed because of empirical evidence of its shortcomings; however, these changes are not yet structural, so the housing market in Chile offers a unique case in terms of reviewing supply, demand and price formation relationships without state regulation (Boano & Perucich, 2017; Perucich, 2019). Precisely because of the absence of state regulation, the explanatory variables on housing prices may be due to factors exogenous to the productive process of the real estate market, because macroeconomic factors impact the domestic economy without many obstacles. For example, in 2004 Leung argued that at that time the literature on property price dynamics was scarce and unsatisfactory, suggesting that it was important to move towards interwined explanations between macroeconomic factors and house price cycles (Leung, 2004). With the 2008 subprime crisis, the literature began to study house price cycles in relation to macroeconomic variables in greater depth, broadening this field of research. Liu et al. applying a Gordon growth model indicate that there is a relatively significant covariance between growth expectations and house prices, suggesting that bubbles, i.e. price speculation, could be considered as a key driver of house prices for the Chinese case (Liu et al., 2017). In a cointegration study for the Sydney case, Al-Masum and Lee indicate that domestic market fundamentals had statistically significant relationships with house prices in the long run (Al-Masum & Lee, 2019). Complementarily, Bangura and Lee in 2022 will indicate that these cointegration processes in Sydney depend on each submarket within Greater Sydney (Bangura & Lee, 2022).  For the case of Chile, this type of causal modelling is absent in the literature.

The discussion of the variable selection is scant. No literature has been provided to justify the use of these variables to examine the causality relationship between housing prices and financial factors.  Why these variables have been selected? What are the theoretical frameworks can be used to justify the use of these variables?

Response: Thanks for this observation, however, there is literature provided to inform the selection of variables in the introduction and also in the methodology. I quote the sections: between line 45 and 92 in the introduction and 109 and 128 in the methodology section. Both sections have references and literature to inform the decisions. It may be expanded but the statement of no literature to justify the use of these variables would require further details, please.

In addition, the use of Santiago de Chile as the dataset for this study. I am wondering whether the use of the disaggregated data. This is a housing submarket approach instead of national level. The housing submarkets should also be discussed. Again, extensive studies of housing submarkets have not been included in the literature.

Response: Thank you for your comment. A justification of the case has been added to the methodology section as indicated below:

To study the Chilean case, we specifically review the submarket of Greater Santiago, the nation's main metropolitan area that concentrates 79% of the housing market supply and 40% of the national population. Wu and Sharma point out the advantages of analysing a submarket based on the presence of continuous and interdependent spatial divisions that allow for a comparative analysis of spatio-temporal changes in the housing markets under investigation (Wu & Sharma, 2012). The case of Santiago de Chile, in addition to concentrating most of the national housing market, is composed of 32 interdependent communes with high levels of socioeconomic segregation and spatial division of functions (Correa-Parra et al., 2020; Vergara-Perucich et al., 2020) that allow us to meet the criteria suggested by Wu and Sharma to enrich housing market studies.

The methodology should be discussed further. The first issue should be the use of weekly data. This is an advantage of this study by using a high-frequency data. This should be discussed more clearly why the authors use the weekly data instead of other frequency. Although this is an advantage, the authors should discuss whether some data (e.g. central bank interest rate) fluctuate so frequently (weekly). Some studies have discussed the use of daily or weekly data in housing.

Response: The comment is gratefully acknowledged and a more detailed description of the reasoning behind the periods and seasonality chosen for the modelling has been incorporated:

In this research it has been decided to work with weekly data by moving averages after cleaning the daily database of outliers and missing data. It has also been sought to clean the data leaving only observations from Monday to Friday. This has been applied since some explanatory variables did not have daily series because they did not incorporate information from weekends and holidays. In order to unify the sample, this strategy of normalisation by weeks is carried out, resulting in an initial total of 520 weeks of continuous time series analysis. Finally, the database is smoothed by moving averages every 3 weeks, as suggested by Hamilton (2018). The initial time series runs from September 2009 to September 2019. These dates arise from two main criteria: availability of detailed data for all the chosen variable series and because in September 2009 the effect of the subprime crisis had already subsided in Chile and in October 2019 Chile enters a process of deep social crisis that distorts the housing market as a result of a series of riots and stoppages of productive activities at different levels (Arias-Loyola, 2021).

  1. The stationary test is up to 15 lags. This equals to a lag of 3-4 months. Some explanation is required why 15 lags were selected. In addition, the authors should consider KPSS as an alternative stationary test to confirm the stationary of the variables as these are so critical steps for any Granger causality test.

Response: The observation is appreciated, although this is explained by the application of the Dickey-Fuller test and the use of the Akaike criterion indicated in table 3. The paragraph preceding the table explains the application of the test and the criteria used to decide how many lags to use in the causality test.

  1. The results can be further discussed. This can be expanded. One of enhancing the discussion is to refer the results to the previous empirical findings. For instance, mortgage interest rate causality results in this study. Is it like previous findings from Al-Masum and Lee (2020) in IJHMA and Liu et al. (2017) in JREFE? If not, then why not. This could be attributed to the importance of international evidence.

Response: Thanks for these comments, a brief reflection has been incorporated in the discussion section considering the specificity of the Chilean case in relation to Sydney. The case of Liu et al. was included in the introduction, but the methods and findings are not related to this study so they were not included in the discussion section.

These results converse with similar findings in the Sydney case by Al-Masum and Lee (2020), who found causal relationships measured by cointegration between house prices and macroeconomic variables such as unemployment, GDP and household income. However, Al-Masum and Lee's methodology is different, so these results are valuable, given that two different cases with different methodologies give validity to the causal relationship between macroeconomic variables and house prices.

Minor comments:

  • Proof reading is required. Numerous editorial mistakes have been identified.

2) Table 4- please use the granger causality coefficient as well.

3) Please follow the recommended format by Economies. This is crucial to ensure that the paper follows the recommended formats.

Response: Thanks, these minor comments has been addressed in the new version of the manuscript.

Reviewer 2 Report

Thank you very much for having a chance to review the paper. Please find my comments below:

  1. There is no detailed review of the literature on the phenomenon of financialization and the methods of measuring this phenomenon, mainly in relation to the real estate market.
  2. The choice of research methods requires more explanation. Critical discussion of the chosen method is missing. Moreover, the authors should provide an explanation of why they applied the selected methods. Are these methods the best fit for the empirical data?
  3. Why does the data only cover the period only until January 2018? The choice of the time span of the research should be explained
  4. The Granger causality test can be conducted separately according to different cases of series stationary: (1) VAR based Granger causality test when both X and Y are stationary, (2) VEC based Granger causality test when both X and Y are not stationary but co-integrated, (3) VAR based Granger causality test for differentiate series when both X and Y are not stationary and not co-integrated. Based on the ADF test it is shown that the differentiate series are stationary. I don't see the results of the cointegration test. Did the authors apply VAR based or VEC Granger causality test? I cannot see the VAR/VEC model results.
  5. In the conclusions, the authors wrote, "For future studies, the data used are suitable for developing other types of time series 291 analysis, such as ARIMA models or especially analysis with VAR models." The Granger test is based on the VAR or VECM models. Thus, is it the plan for future study? It raises doubts about whether the research is reliable and robust.
  6. There is a lack of limitation of the study in the conclusion section.

Author Response

RESPONSE TO REVIEWERS ECONOMIES

PAPER: Is there Financialization of Housing Prices? Empirical Evidence from Santiago de Chile

economies-1692374

Thanks to the editors of Economies for allowing me to amend the paper following the suggestions of reviewers. I did a comprehensive revision of the article, by considering the valuable contributions of reviewers. Both provided substantial observations to the paper giving precise details on how to improve it. This make it easier to address the revisions required and to improve it.

Based on these revisions, I understand the point of both reviewers was to request deeper reflections taking advantage of the data produced in the research process. Although these revisions demanded more work and re-write entire paragraphs and even new calculations, I believe the result is much better than in the original version.

Here are the responses for each requirement and/or comments made by the reviewers.

REVIEWER 2

There is no detailed review of the literature on the phenomenon of financialization and the methods of measuring this phenomenon, mainly in relation to the real estate market.

Response: The comment is appreciated, although the observation is not understood considering that from line 26 to line 82 of the originally submitted manuscript arguments are elaborated from concepts from an extensive literature review on the process of housing financialisation.

The choice of research methods requires more explanation. Critical discussion of the chosen method is missing. Moreover, the authors should provide an explanation of why they applied the selected methods. Are these methods the best fit for the empirical data?

Response: The comment is appreciated, however from line 138 to 150 of the original manuscript the criteria for using a Granger causality test is explained. Other articles have been reviewed to try to understand the observation and none of them elaborate much more than what is already elaborated in this manuscript on why to use this technique and not another. The wording is revised to elaborate on the reasons for using this method.

Why does the data only cover the period only until January 2018? The choice of the time span of the research should be explained

Response: Thanks for the precise observations. We applied a grammar revision and edition by a professional external service to address the language flaws. Given that English is not out mother tongue, of course, a final revision by the editors would be great too.

Thank you for this observation, which was certainly an error in the text, which was remedied by the following paragraph included in the methodology section:

In this research it has been decided to work with weekly data by moving averages after cleaning the daily database of outliers and missing data. It has also been sought to clean the data leaving only observations from Monday to Friday. This has been applied since some explanatory variables did not have daily series because they did not incorporate information from weekends and holidays. In order to unify the sample, this strategy of normalisation by weeks is carried out, resulting in an initial total of 520 weeks of continuous time series analysis. Finally, the database is smoothed by moving averages every 3 weeks, as suggested by Hamilton (2018). The initial time series runs from September 2009 to September 2019. These dates arise from two main criteria: availability of detailed data for all the chosen variable series and because in September 2009 the effect of the subprime crisis had already subsided in Chile and in October 2019 Chile enters a process of deep social crisis that distorts the housing market as a result of a series of riots and stoppages of productive activities at different levels (Arias-Loyola, 2021).

The Granger causality test can be conducted separately according to different cases of series stationary: (1) VAR based Granger causality test when both X and Y are stationary, (2) VEC based Granger causality test when both X and Y are not stationary but co-integrated, (3) VAR based Granger causality test for differentiate series when both X and Y are not stationary and not co-integrated. Based on the ADF test it is shown that the differentiate series are stationary. I don't see the results of the cointegration test. Did the authors apply VAR based or VEC Granger causality test? I cannot see the VAR/VEC model results.

Response: The comment is appreciated. A table with the results of the cointegration test is added and it is clarified that the test is based on a VAR model and not on a VEC.

To confirm the robustness of the selected dataset after transformation, for the order of 15 lags to be reviewed, an Engle-Granger cointegration test is applied, which seeks to identify whether a pair of variables is cointegrated by analysing the residuals of a regression between variables to identify by Augmented Dickey-Fuller test whether a unit root exists. Table 4 indicates the results confirming the robustness of the variables to apply a Granger causality test.

Table 4. Engle-Granger cointegration test results.

Variables on Housing Prices

lags

Engle-Granger Coefficient

p-value

IPSA

15

-7.1113275

0.01

Monetary Policy Rate

15

-6.8507553

0.01

Cash Circulation

15

-7.6780385

0.01

Inflation

15

-7.5782503

0.01

Mortgage Interest Rate

15

-7.6728005

0.01

CSI 300

15

-7.7215235

0.01

FTSE 100

15

-7.8595879

0.01

Dow Jones

15

-7.6870395

0.01

In the conclusions, the authors wrote, "For future studies, the data used are suitable for developing other types of time series 291 analysis, such as ARIMA models or especially analysis with VAR models." The Granger test is based on the VAR or VECM models. Thus, is it the plan for future study? It raises doubts about whether the research is reliable and robust.

Response: It is understood that there is a criticism underlying the study that is pertinent: Why not incorporate in this same study a VAR modelling between the variables? In this case, only a Granger causality test has been applied following the methodology applied by other indexed articles that also use this method, published in high impact journals. It is clarified in the methodology that this research focuses on a Granger causality test by VAR model, but it is not an autoregressive vector analysis. This, together with the incorporation of a cointegration test and different clarifications requested by both reviews, clarifies the reliability of the analysis.

There is a lack of limitation of the study in the conclusion section.

Response: The observation is acknowledged and a detailed account of the limitations of this study, which largely stem from the constructive observations made by the review, is incorporated.

The study has some limitations. Firstly, it focuses on the case of Greater Santiago, which although it concentrates a large part of the Chilean housing market, at the level of other cities the results could vary. That is, financialisation processes could affect medium-sized cities and regional capitals differently. However, data at the level of these cities are scarcer and at the moment more difficult to model. It is a pending task for further studies to run these causality models at the national level. Another limitation is that there are other explanatory factors on housing prices that are not macroeconomic but that could help to make the analysis more complex, such as the price of construction materials, land prices or the socio-economic characteristics of the neighbourhood. Incorporating these variables could allow us to identify whether financial and macroeconomic factors have a greater statistical weight on house prices than other fundamentals identified in the literature. Finally, a second derivative from the dataset used for this study could develop an autoregressive vector analysis to look for the impact that certain variables have on house prices in the event of a shock to any of them. The latter is rather an opportunity to deepen these results.

s, these minor comments has been addressed in the new version

Round 2

Reviewer 1 Report

This revised version has been significantly improved. The paper incorporated many of my previously mentioned comments satisfactorily. I can see the improvement. I do enjoy reading the revised paper that see most comments have been considered. However, I do have some minor comments that can further improve the paper.

  • I do appreciate that the literature review has been extended. However, I reckon that the study should try to include more previous studies, particularly in emerging markets, particularly in Chile. Is it the first study in Chile? How this study differs from previous Chilean studies, if any? For instance, Vergara-Peruich and Aguirre-Nunez (2019) in Buildings.

Vergara-Perucich, J. F., & Aguirre-Nuñez, C. (2019). Housing prices in unregulated markets: Study on verticalised dwellings in Santiago de Chile. Buildings10(1), 6.

  • The contribution of this study should not be explicitly discussed in the introduction. How this study contributes to the literature.
  • Page 3- I acknowledge that the literature of housing submarkets has been discussed. However, I reckon that the authors can further discuss the importance of housing submarkets consideration. The clear socio-economic and demographic disparities across the regions of the city have been widely documented in developed and emerging markets (see Randolph and Tice, 2014; Wang and Lee, 2022). The ignorance of housing submarkets could be the possible cause of the mixed results in the previous housing studies (Costello et al., 2011). Given this has been considered by the authors, why not sing this loud? I think this would be a contribution of this study if no previous Chilean studies consider this aspect previously. Please have a stronger component on this.

Randolph, B. & Tice, A. (2014) Suburbanizing disadvantage in Australian cities: sociospatial change in an era of neoliberalism, Journal of Urban Affairs, 36, pp. 384–395.

Wang, J., & Lee, C. L. (2022). The value of air quality in housing markets: A comparative study of housing sale and rental markets in China. Energy Policy160, 112601.

Costello, G., Fraser, P. & Groenewold, N. (2011) House prices, non-fundamental components and interstate spillovers: the Australian experience, Journal of Banking & Finance, 35, pp. 653–669.

  • The frequency of data can be further discussed. Again, the study does have a high-frequency dataset. This should be highlighted. Unfortunately, this is not the case. Higher frequency data offers extra insights that has been largely ignored by the low-frequency data (see Cotter and Stevenson, 2006; Lee, Stevenson and Lee, 2018) in the property market. Can the Spline-GARCH be considered to address the issues that have been discussed in page 4 (different frequency datasets) as discussed previously? If not, some qualifications should be provided.

Cotter, J., & Stevenson, S. (2008). Modeling long memory in REITs. Real Estate Economics, 36(3), 533-554.

Lee, C. L., Stevenson, S., & Lee, M. L. (2018). Low‐frequency volatility of real estate securities and macroeconomic risk. Accounting & Finance, 58, 311-342.

  • Please follow the suggested format, and the reference list has not been updated in the revised version.
  • Proof-reading is required. 

Author Response

RESPONSE TO REVIEWERS ECONOMIES

Second Round: Minor Revisions

PAPER: Is there Financialization of Housing Prices? Empirical Evidence from Santiago de Chile

economies-1692374

Thanks to the editors and referees of Economies for sharing this second round of observations to improve the article. Only one reviewer provided new insights to amend in the article. Here are the responses for each requirement and/or comments made by the reviewers.

REVIEWER 1

  1. I do appreciate that the literature review has been extended. However, I reckon that the study should try to include more previous studies, particularly in emerging markets, particularl in Chile. Is it the first study in Chile? How this study differs from previous Chilean studies, if any? For instance, Vergara-Peruich and Aguirre-Nunez (2019) in Buildings.

Vergara-Perucich, J. F., & Aguirre-Nuñez, C. (2019). Housing prices in unregulated markets: Study on verticalised dwellings in Santiago de Chile. Buildings, 10(1), 6.

Response: Thanks for this comments, however the requested references and contents are already in the paper, between line 69 and 92.

  1. The contribution of this study should not be explicitly discussed in the introduction. How this study contributes to the literature.

Response: I guess the comment was that the contribution of the paper should be explicitly discussed in the literature.

A new paragraph was included highlighting the value of this contribution to literature, placing the paper among other articles that have also worked with causal models for housing prices.

There is literature that uses the Granger causality test to analyse house prices, but none has been developed on the housing submarket in Chile. Aye (2018) conducted a Granger causality analysis to see if economic policy uncertainty had an effect on housing prices, but grouping macroeconomic variables at the level of 8 nations including Chile, but not with a specific focus on the Chilean housing market. Hong and Li conducted a Granger causality analysis between housing prices and the willingness to invest of financial actors for the case of China, however, there are no other articles prior to this one that evaluate the causal relationship from the theoretical framework of financialisation with housing prices, neither for the case of Chile nor for other cases, as far as this research project has been able to review. This makes the results presented here of great value to think tanks in the field of financialisation, housing and real estate markets.  

  1. Page 3- I acknowledge that the literature of housing submarkets has been discussed. However, I reckon that the authors can further discuss the importance of housing submarkets consideration. The clear socio-economic and demographic disparities across the regions of the city have been widely documented in developed and emerging markets (see Randolph and Tice, 2014; Wang and Lee, 2022). The ignorance of housing submarkets could be the possible cause of the mixed results in the previous housing studies (Costello et al., 2011). Given this has been considered by the authors, why not sing this loud? I think this would be a contribution of this study if no previous Chilean studies consider this aspect previously. Please have a stronger component on this.

Response: Thanks for this recommendation,  a new paragraph has been included making reference to the quite interesting papers suggested.

Santiago is a highly segregated city (Rasse, 2016), so there are significant disparities in the different areas of the metropolis, a situation that has an impact on the housing market for emerging economies (Randolph & Tice 2014; Wang & Lee, 2022). In addition, Chile is a nation with very unequal cities (Vergara-Perucich, Urbano, 2021), so the consideration of Santiago implied defining this metropolitan space as a specific submarket. In several previous studies, the spatial qualities and organisations of housing markets are not considered methodologically and therefore can generate results that are difficult to generalise as indicated by Costello et al. (2011). In this case, being aware of the high segregation of the housing market in Greater Santiago and the inequality among Chilean cities, an ad-hoc methodological design was considered necessary, given that the macroeconomic variables included in this study would possibly not have the same effect in smaller cities or cities dependent on other productive activities within the same nation. For example, Idrovo-Aguirre and Contreras-Reyes have identified that cities in northern Chile are more sensitive to changes in the price of copper, given that these cities are highly dependent on this commodity (Idrovo-Aguirre & Contreras-Reyes, 2021). In the case of Santiago, the main economic dependence is on financial services. 

  1. The frequency of data can be further discussed. Again, the study does have a high-frequency dataset. This should be highlighted. Unfortunately, this is not the case. Higher frequency data offers extra insights that has been largely ignored by the low-frequency data (see Cotter and Stevenson, 2006; Lee, Stevenson and Lee, 2018) in the property market. Can the Spline-GARCH be considered to address the issues that have been discussed in page 4 (different frequency datasets) as discussed previously? If not, some qualifications should be provided.

Response: Thanks for the recommendation, a new paragraph was included in the description of data echoing this observation.

By using time series with weekly observations, a high-frequency data set is achieved, which reduces the statistical noise that can be generated by lower frequency series that do not recognise the volatility of financial markets or investment risk (Lee et al. 2018), which for real estate analysis offers a greater ability to accurately study long-term variations (Cotter & Stevenson, 2008).

  1. Please follow the suggested format, and the reference list has not been updated in the revised version.

Response: Thanks for the observation, the paper was adjusted and special care was taken with the reference list.

  1. Proof-reading is required.

Response: Thanks, a throughout revision was applied to the text.

Reviewer 2 Report

The authors addressed all my comments. I accept the paper in its present form.

Author Response

Thanks for taking the time in reviewing my work and for your previous comments.

Kinds regards.